# Salvage Partial Laryngectomy after Failed Radiotherapy: Oncological and Functional Outcomes

**DOI:** 10.3390/jcm11185411

**Published:** 2022-09-15

**Authors:** Mélanie Gigot, Antoine Digonnet, Alexandra Rodriguez, Jerome R. Lechien

**Affiliations:** 1Department of Otorhinolaryngology and Head and Neck Surgery, CHU Saint-Pierre, 1000 Brussels, Belgium; 2Department of Surgery, Bordet Institute, Brussel Free University, 1050 Brussels, Belgium; 3Department of Otolaryngology, Polyclinic of Poitiers, Elsan, 86000 Poitiers, France; 4Department of Human Anatomy and Experimental Oncology, Faculty of Medicine, UMONS Research Institute for Health Sciences and Technology, University of Mons, 7000 Mons, Belgium; 5Department of Otolaryngology-Head & Neck Surgery, Foch Hospital, School of Medicine, UFR Simone Veil, Université Versailles Saint-Quentin-en-Yvelines, 92150 Paris, France

**Keywords:** otolaryngology, head neck surgery, laryngectomy, partial, cancer, oncological, survival, voice, swallowing

## Abstract

Objective: To investigate oncological and functional outcomes in patients treated with salvage partial laryngectomy (SPL) after failed radio/chemotherapy. Study design: Retrospective multicenter chart review. Methods: Medical records of patients treated with SPL from January 1998 to January 2018 in two University Medical centers were retrieved. The SPL included horizontal supraglottic laryngectomy, hemi-laryngectomy and crico-hyoido-epiglottopexy. The following outcomes were investigated: histopathological features; overall survival (OS); recurrence-free survival (RFS) local and regional controls; post-operative speech recovery; and the oral diet restart and decannulation. Results: The data of 20 patients with cT1–cT3 laryngeal cancer were collected. The mean follow-up of patients was 69.7 months. The mean hospital stay was 43.0 days (16–111). The following complications occurred in the immediate post-operative follow-up: neck fistula (N = 6), aspiration pneumonia (N = 5), and chondronecrosis (N = 2). Early or late total laryngectomy was carried out over the follow-up period for the following reasons: positive margins and local recurrence/progression (N = 7), chondronecrosis (N = 2) and non-functional larynx (N = 1). The restart of the oral diet was carried out in 12/15 (80%) SPL patients (five patients being excluded for totalization). All patients recovered speech, and decannulation was performed in 14 patients (93%). The 5-year OS and RFS were 50% and 56%, respectively. The 5-year local and regional control rates were 56% and 56%, respectively. Conclusions: Partial laryngectomy is an alternative therapeutic approach to total laryngectomy in patients with a history of failed radiation.

## 1. Introduction

Head and neck squamous cell carcinoma (HNSCC) is the 6th most common adult malignancy worldwide, accounting for 5.3% of all cancers [1]. Laryngeal squamous cell carcinoma (LSCC) is the second most prevalent HNSCC, corresponding to 211,000 new cases and 126,000 deaths per year worldwide [2,3]. According to the stage and location of tumor, the main therapeutic options include surgery, radiotherapy or concurrent chemoradiotherapy. Radiotherapy may achieve local control rates of 77% to 100% of cT1, 62% to 83% of cT2, and 50% to 76% of cT3 supraglottic SCCs [4]. In cases of local failure, the most common therapeutic option remains total laryngectomy, which may be associated with complications and poor quality of life outcomes [5]. However, a recent systematic review of the literature suggested that the realization of salvage partial laryngectomy (SPL) in place of total laryngectomy may be an option in selected cases of glottic or supraglottic SCC recurrence or post-radiotherapy failure, reporting adequate local control and survival outcomes [6].

In this study, we retrospectively reviewed the oncological, histopathological and functional outcomes of patients who benefited from SPL after failed radiotherapy.

## 2. Materials and Methods

The data of patients treated with post-radiotherapy SPL for glottic or supraglottic SCC from 1998 to 2021 were retrieved. Patients were recruited and treated in two University medical centers (CHU Saint-Pierre and Jules Bordet Institute, Brussels, Belgium). The SCCs were located in glottic or supraglottic region. The SPL indication was based on cTNM tumor (cT1–cT3), medical history of patients, the MRI/CT-scan findings and swallowing tests. All patients underwent diagnostic preoperative workup with laryngeal fibroscopy, in-suspension laryngoscopy (biopsy) and injected tomodensitometry or Pet-CT. The SPL was considered as a therapeutic option if it was possible to preserve one cricoarytenoid unit after the surgery. The SPL was not recommended for the following patient/tumor outcomes: cricoid invasion; large invasion of laryngeal posterior commissure or thyroid cartilage; and pharyngeal or prevertebral invasion (cT4). The SPL decision was approved by the multidisciplinary oncological board. The local institutional review boards of both centers approved the study design (CE171211 et CE2849).

### 2.1. Procedures

Surgeries were performed under general anesthesia. According to the European Laryngological Society classification [7], the following SPL were considered: open partial horizontal supraglottic (hemi) laryngectomy (type I), supracricoid laryngectomy with cricohyoidoepiglottopexy (type IIa) or cricohyoidopexy (type IIb; Appendix A). Note that in all procedures, surgeons performed extemporaneous analyses. In the case of positive margins, the surgeon completed the surgery with recut of the positive tissue section. We did not extend the resection from one type of open partial laryngectomy to another. Only the definitive margin analyses were considered for the follow-up decisions. The surgical techniques were described in previous studies [7]. According to the tomodensitometry findings, unilateral or bilateral selective neck dissection(s) of levels II-IV were carried out in cT3 and selected cT2 patients.

Patients had tracheostomies and feeding tubes. All patients benefited from post-operative speech therapy (3- to 5 sessions weekly), which was started 7 days after the surgery. All patients received 7-day antibiotics and 1-month proton pump inhibitors. An oral diet was restarted from 7 days post-SPL depending on the speech therapist and otolaryngologist’s agreement. Regarding the evolution of swallowing, speech and breathing, patients were decannulated and discharged as soon as possible after a fiberoptic endoscopic evaluation of swallowing.

The total laryngectomy was considered in patients with (i) positive margins at the post-SPL histopathological analysis, (ii) post-operative chondronecrosis or non-functional larynx, or (iii) for local recurrence. If the salvage total laryngectomy was performed in the 6-month post-SPL follow-up, the total laryngectomy was considered as an early procedure, while when the surgeon carried out the total laryngectomy after 6 months of follow-up, the salvage laryngectomy was considered as late. Note that positive margins (R1) were followed before the decision of total laryngectomy according to the difficulties to have reliable histopathological findings in radiation tissues.

The neck dissection was proposed by the multidisciplinary oncological team in patients with suspicion of positive nodes at the tomodensitometry even if the cTNM classification reported cN0 (<1 cm diameter). Unilateral neck dissection was proposed for suspicion of unilateral node(s), while bilateral neck dissection was proposed for suspected bilateral nodes.

### 2.2. Outcomes

The following postoperative outcomes were retrieved: infectious complications; fistula; chondronecrosis; aspiration pneumonia; death. The functional outcomes included the post-operative speech recovery; restart of an oral diet; and the removal of the tracheostomy tube. Post-operative speech recovery was defined as the ability of the patient to be understood by other people without repetition. The usual voice and speech clinical tools (e.g., speech rate; GRBASI, voice handicap index) were not used because they are inappropriate for post-partial laryngectomy voice and speech evaluations and there was no systematic use of clinical voice/speech tools in our centers. The following oncological outcomes were considered: 3- and 5-year overall survival (OS); recurrence-free survival (RFS); and local and regional control rates.

## 3. Results

### 3.1. Setting and Patients

Twenty patients underwent SPL after failed radiotherapy (N = 17) or concurrent chemoradiotherapy (N = 3), including supracricoid laryngectomy with cricohyoidoepiglottopexy (type 2a, N = 11); open partial horizontal supraglottic laryngectomy (type 1, N = 6), and open partial horizontal hemi-laryngectomy (type 1, N = 3). The flow chart of the study is described in Figure 1.

There were 14 males and the mean age was 54.0 years old. Eighteen patients (90%) reported a history of tobacco consumption prior to the initial radiation treatment, while 14/16 (88%) had a history of chronic alcohol consumption (>3 IU/day). The radiotherapy doses of initial treatment are reported in Appendix A. Two patients had chemoradiation prior to salvage surgery (cisplatin). The surgeons performed six unilateral and five bilateral neck dissections. There were 7, 10, and 3 cT1, cT2, and cT3N0M0 glottic or supraglottic SCC, respectively (Table 1).

Synchronous cancer was detected in two patients (esophagus and lung cancer).

The histopathological findings are reported in Table 1. Frozen sections were positive in 7 cases, leading to preoperative revision. Among them, four definitive histopathological examinations reported R1, which led to a follow-up approach and total laryngectomy for recurrence (Appendix A). Three of eleven neck dissections were positive. Among them, two patients had extra-capsular node invasion; which led to post-operative chemo-radiotherapy and palliative chemotherapy, respectively. As described in Appendix A, the initial cTNM assessment was lower than the pTNM for five patients with two with pT4 cancer at the histopathological examination.

The mean hospital stay was 43.0 days (16–111). The following complications occurred in the immediate post-operative follow-up: neck fistula (N = 6), aspiration pneumonia (N = 5), and chondronecrosis (N = 2). There was no local infection (abscess) or death during the post-operative course. Three patients with fistula required re-intervention and the following flaps were used: pectoral (N = 1), supraclavicular (N = 1) and temporal (N = 1) flap. Two patients with chondronecrosis, one patient with positive margins and rapid (<6-month) progression (R1) and one patient with non-functional larynx were treated with an early total laryngectomy (<6-month post-SPL). Late total laryngectomy (>6 months of follow-up) was carried out in six patients (Figure 1). Two had positive margins but refused the total laryngectomy and three had late recurrences. Finally, total laryngectomy was carried out in 10 patients (50%) for the following reasons: positive margins and follow-up recurrence/progression (N = 7); chondronecrosis (N = 2); and non-functional larynx (N = 1; Appendix A).

### 3.2. Functional Outcomes

The functional outcomes are summarized in Table 1. The restart of an oral diet was carried out in 12/15 (80%) SLP patients without early total laryngectomy with a mean delay of 49.0 days. Three patients did not restart an oral diet because of severe aspiration (N = 2) or malnutrition related to synchronous chest cancer. These patients benefited from permanent gastrostomy. The speech rehabilitation was successfully completed in all patients with a mean delay of 60.7 days (mean of speech session: three times weekly). Decannulation was carried out in 14 patients (93%) with a mean delay of 11.2 days. The restart of oral diet, speech rehabilitation and decannulation delays varied between surgeries (Table 1).

### 3.3. Oncological Outcomes

The mean follow-up of patients was 69.7 months (6–178). The 3- and 5-year OS were 50.0% and 50.0%, respectively. The 3- and 5-year local control rates were 55.5% and 55.5%, respectively. The 3- and 5-year RFS were 55.5% and 55.5%, respectively. At the end of the follow-up, five patients were alive and five were lost of follow-up (after the 5-year initial follow-up). The causes of death of the 10 remaining patients were local recurrence or distant metastasis (N = 5); non-oncological origin (N = 3); and metachronous non-head and neck cancer (N = 2). Note that 3/6 patients who underwent salvage total laryngectomy were alive at the end of the follow-up period (five patients were lost of follow-up). In the SPL group, 5/10 patients were alive at the end of the follow-up period (Appendix A).

## 4. Discussion

Residual or recurrent LSCC after failed radiotherapy is a challenging issue. The salvage total laryngectomy was the main option for post-radiation LSCC and only a few case series with a low number of patients investigated the oncological and functional outcomes of SLP [6].

The primary finding of the present study was the demonstration of adequate functional post-operative outcomes, including decannulation, speech rehabilitation and oral diet restart. All patients with SPL were successfully decannulated after 60 days. Our decannulation rate was slightly above that of those found in the literature despite a longer delay of decannulation in the present study [8,9,10,11,12]. The oral restart was possible in 80% of patients with a mean delay of 49 days. Kim et al. reported oral restart rates of 100% versus 74.2% in patients who underwent salvage supraglottic laryngectomy or TL, respectively [12]. The mean removal time of the feeding tube was 25 days in the study of Kim et al., which was, however, substantially shorter than our delay [12]. In the study of Philippe et al., 15/20 (75%) patients treated with SPL for LSCC after failed radiotherapy were able to restart an oral diet in the post-operative few months [11]. Others reported similar rates of the restart of an oral diet with delays ranging from 15 to 74 days [6,8,9,10,12,13]. Our study reports an adequate post-operative speech rehabilitation rate despite longer delays than those of some studies in which authors started speech exercises 3 days after the SLP [6,10]. The functional outcome in comparison with other studies is still limited regarding the heterogeneity across studies about the types of SPL, the patient comorbidities, and the TNM features. Moreover, as supported by Paleri et al., the differences between world regions in speech therapy access and program may support different speech rehabilitation outcomes [14].

The complications after salvage partial laryngectomy depend on the type of surgery and the features of the population (comorbidities), and include most commonly fistula, hemorrhage, wound infection, aspiration pneumonia and dysphagia [10,11,12,13,14,15,16,17]. In this case series, aspiration pneumonia (25%), fistula (15%) and chondronecrosis (10%) were the main complications. The rate of fistula in SPL patients after failed radiotherapy ranged from 2% to 81% [15,16,17,18] and was influenced by the radiotherapy doses rather than the type of surgery [15]. The tobacco and the reflux histories are additional contributing factors to fistula [19]. In these case series, Philippe et al. carried out total laryngectomy in two patients (10%) with a non-functional larynx and recurrent aspiration pneumonia, which corroborates our rate of a non-functional larynx [11]. Aspiration pneumonia occurred in 20% of our patients. The literature rate of aspirations ranges from 3% to 40% [8,11,20,21].

In the present study, the 5-year OS and RFS were 50% and 55%, respectively. The OS and RFS data in the literature substantially vary from one study to another. Overall, both OS and RFS ranged from 52% to 95% but depend on the features of patients (comorbidities), tumor stage and treatment [10,11,16,17,21,22,23]. According to Kim et al., the OS and DFR were non-significantly higher in the salvage SPL than in the salvage TL [12]. Authors reported 5-year OS and RFS of 87.5% and 41.9% in their SPL patients, which were significantly higher for OS than our rate. Interestingly, they highlighted the importance of margin status in survival and recurrence outcomes, which supports the need for a close follow-up in patients with post-SLP positive margins or re-intervention. In the study of Makaieff et al., the 5-year OS was 69% [20], while Philippe et al. reported a 3-year OS of 66% [11]. As for other studies of the literature [9,10,15], the main cohort difference between our case series and these studies was the inclusion of cT3 LSCC in the present study, which is known to be associated with poorer OS and RFS data [15]. Interestingly, our study reported that the initial cTNM assessment may be biased according to the radiation history. Indeed, for six patients, the pTNM was higher than the cTNM, both having a T4 LSCC at the histopathological examination. The tissue fibrosis related to radiation may influence the clinical and imaging staging leading to the inclusion of patients with more advanced disease. This point is an additional factor supporting the low but literature-comparable OS and RFS rates.

The main limitations of the present study were the retrospective design and the low number of patients. However, the SPL after failed radiotherapy remains a rare surgical approach because the current trend in head and neck oncology is to propose salvage total laryngectomy for patients with such LSCC. However, since the possible risk of conversion in total laryngectomy exists, appropriate information about the patient is a mainstay in this therapeutic approach.

## 5. Conclusions

The salvage partial laryngectomy after radiotherapy failure is an alternative therapeutic option to total laryngectomy for patients with cT1-T3 LSCC. Otolaryngologists had to be careful about the risk of preoperative mis-staging. Aspiration and fistula were the most common complications occurring in 15% to 20% of cases.

## Figures and Tables

**Figure 1 jcm-11-05411-f001:**
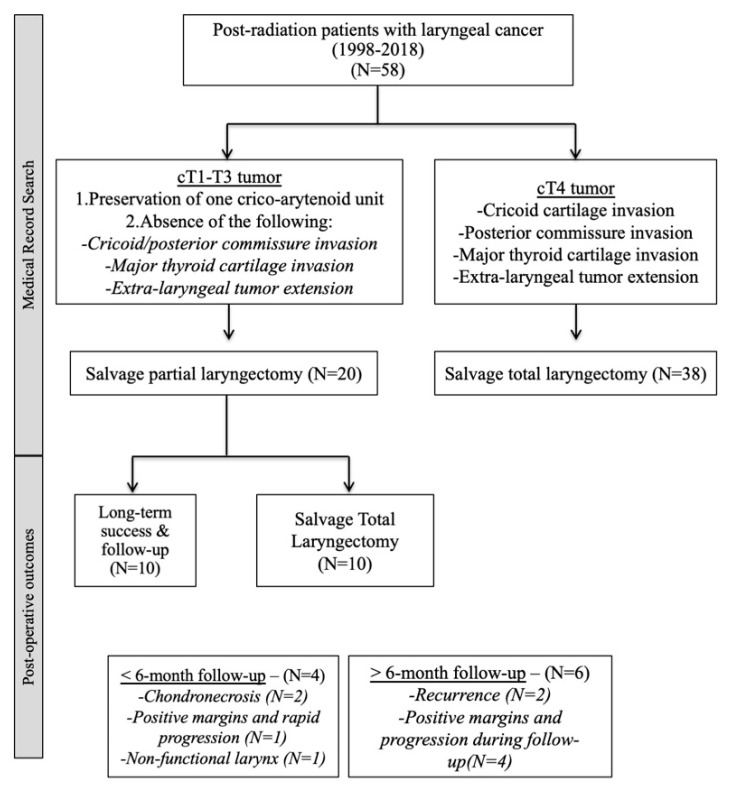
Flow chart.

**Table 1 jcm-11-05411-t001:** Epidemiological, clinical and surgical features.

**Outcomes**	**Mean/N**
Age (mean, range)	54 (40–69)
Gender (F/M)	6/14
Tumor features	
Second location (N, mean month delay)	4 (52.5)
Recurrence (N, mean month delay)	16 (34.8)
Locations	
Supraglottic	7 (35)
Glottic	10 (50)
Laryngopharyngeal	3 (15)
Stages	
cT1N0M0	7 (35)
cT2N0M0	10 (50)
cT3N0M0	3 (15)
Surgeries	
Cricohyoidoepiglottopexy	11 (55)
Horizontal supraglottic laryngectomy	6 (30)
Horizontal hemi-laryngectomy	3 (15)
Tracheostomy	20 (100)
Feeding tube	20 (100)
**Histopathological Findings**	**N (%)**
Margins	
R0	13 (65)
R1	7 (35)
Management of R1	
Re-intervention	5 (71)
Total laryngectomy	2 (29)
Neck dissection	
pN+	3 (27)
pN0	8 (73)
Management of N+	
Follow-up	1 (33)
Chemoradiotherapy	1 (33)
Palliative chemotherapy	1 (33)
**Functional Outcomes**	**N (%)**
Early total laryngectomy	5 (25)
Chondronecrosis	2 (10)
Positive margins	2 (10)
Non-functional larynx	1 (5)
Late total laryngectomy	5 (25)
Positive margins	2 (10)
Recurrences	3 (15)
Oral diet rehabilitation	
Success of restart	12 (80)
Definitive gastrostomy	3 (20)
Delay (mean, days)	49.0
Cricohyoidoepiglottopexy	34.3
Horizontal supraglottic laryngectomy	163.0
Horizontal hemi-laryngectomy	58.5
Speech rehabilitation	
Success	15 (100)
Delay (mean, days)	60.7
Cricohyoidoepiglottopexy	52.8
Horizontal supraglottic laryngectomy	90.0
Horizontal hemi-laryngectomy	56.6
Tracheostomy	
Decannulation	14/15 (93)
Delay (mean, days)	11.2
Cricohyoidoepiglottopexy	10.4
Horizontal supraglottic laryngectomy	13.8
Horizontal hemi-laryngectomy	7.6

Abbreviations: F/M = female/male; R = margin status; N = number of cases.

## Data Availability

Data are available on request.

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
