# Peer review of "Salvage Partial Laryngectomy after Failed Radiotherapy: Oncological and Functional Outcomes"

_jcm, 2022, doi:10.3390/jcm11185411_

Round 1

Reviewer 1 Report

This manuscript is about salvage partial laryngectomy for laryngeal cancer. The authors reported that partial laryngectomy would be an alternative therapeutic approach because almost all of patients with partial laryngectomy could success decannulate and oral diet. This manuscript is interesting but there is a big problem about novelty.

Major comments

1.      Although you mentioned few previous report about salvage partial laryngectomy, there has been previous reports including systematic review. I think this methods depend on the skill of surgeon and the status of patients. Therefore, you should compare the standard treatment of salvage total laryngectomy with salvage partial laryngectomy like this paper as below. As a result, the surgeon and patient should choose which treatment is better.

Kim, J. H., Kim, W. S., Koh, Y. W., Kim, S. H., Byeon, H. K., & Choi, E. C. (2018). Oncologic and functional outcomes of salvage supracricoid partial laryngectomy. Acta oto-laryngologica138(12), 1117–1122. https://doi.org/10.1080/00016489.2018.1506154

2.      You should added more details about the patients characteristics such as smoking status, performance status, lung function and RT dose.

3.      You should explain more details about the selection of partial laryngectomy and total laryngectomy. i.e. preoperative age, swalloing examination and performance status. In this study, 10 out of patients were performed total laryngectomy, and I think that is not a small number. Therefore, you should mention future indications.

4.      I can not understand the terms of “late” and “early”. You should mention in material and methods.

5.      You should write details about the contents of the re-intervention.

Minor comments

1.      In table 1, is the number of horizontal hemi-laryngectomy 3?

Author Response

Editor in chief

J Clin Med

Mons, Augustus, 2022

Dear Professor,

I’m sending the revised paper entitled: “Salvage Partial Laryngectomy After failed Radiotherapy: Oncological and Functional Outcomes." (by Gigot et al.) which is submitted for publication in J Clin Med.

I thank the reviewers for the relevant comments. We considered all of them.

Reviewer 1

This manuscript is about salvage partial laryngectomy for laryngeal cancer. The authors reported that partial laryngectomy would be an alternative therapeutic approach because almost all of patients with partial laryngectomy could success decannulate and oral diet. This manuscript is interesting but there is a big problem about novelty.

Major comments

  1. Although you mentioned few previous report about salvage partial laryngectomy, there has been previous reports including systematic review. I think this methods depend on the skill of surgeon and the status of patients. Therefore, you should compare the standard treatment of salvage total laryngectomy with salvage partial laryngectomy like this paper as below. As a result, the surgeon and patient should choose which treatment is better.

Kim, J. H., Kim, W. S., Koh, Y. W., Kim, S. H., Byeon, H. K., & Choi, E. C. (2018). Oncologic and functional outcomes of salvage supracricoid partial laryngectomy. Acta oto-laryngologica138(12), 1117–1122. https://doi.org/10.1080/00016489.2018.1506154

Thank you for the comment. As requested, we compared our data with the data of the suggested paper in the Discussion:, p.6, line 184: “Our decannulation rate was slightly above that those found in the literature despite a longer delay of decannulation in the present study [8-12]. The oral restart was possible in 80% of patients with a mean delay of 49 days. Kim et al. reported oral restart rates of 100% versus 74.2% in patients who underwent salvage supraglottic laryngectomy or TL, respectively [12]. The mean removal time of feeding tube was 25 days in the study of Kim et al, which was however substantially shorter than our delay [12].

About OS and RFS: we also discussed our results with those of Kim et al.: discussion, P.6, line 211: “In the present study, the 5-year OS and RFS were 50% and 55%, respectively. The OS and RFS data in the literature substantially vary from one study to another. Overall, both OS and RFS ranged from 52% to 95% but depend on the features of patients (comorbidities), tumor stage and treatment [10,11,16,17,21-23]. According to Kim et al., the OS and DFR were non-significantly higher in the salvage SPL than in the salvage TL [12]. Authors reported 5-year OS and RFS of 87.5% and 41.9% in their SPL patients, which were significantly higher for OS than our rate. Interestingly, they highlighted the importance of margin status in survival and recurrence outcomes, which supports the need of a close follow-up in patients with post-SLP positive margins or re-intervention.”

  1. You should added more details about the patients characteristics such as smoking status, performance status, lung function and RT dose.

      We added some of these data. Precisely, you can see in Appendix 1 that we added the tobacco status prior to salvage surgery, the alcohol status, the RT doses of initial treatment and the use of chemotherapy in the initial treatment. About lung function, we did not perform lung functional evaluations before laryngeal cancer treatment because this is not recommended in the French Society of Otolaryngology and Head and Neck Surgery. Sorry.

      You can find the new data in Appendix 1, p.7, line 1, column 2, 3, 5, 6.

      In result section: p.3, setting and patient, line 112: “Eighteen patients (90%) reported a history of tobacco consumption prior to the initial radiation treatment, while 14/16 (88%) had a history of chronic alcohol consumption (> 3 IU/day). The radiotherapy doses of initial treatment are reported in Appendix 1. Two patients had chemoradiation prior to salvage surgery (cisplatin).”

  1. You should explain more details about the selection of partial laryngectomy and total laryngectomy. i.e. preoperative age, swalloing examination and performance status. In this study, 10 out of patients were performed total laryngectomy, and I think that is not a small number. Therefore, you should mention future indications.

In fact, all patients underwent partial laryngectomy. The 10 laryngectomies were carried out because positive margins, recurrence or complications (e.g. chondronecrosis).

According to the comment of the reviewer, we specified that in the methods: p.2, line 87: “The total laryngectomy was considered in patients with i) positive margins at the post-SPL histopathological analysis, ii) post-operative chondronecrosis or non-functional larynx, or iii) for local recurrence. If the salvage total laryngectomy was performed in the 6-month post-SPL follow-up, the total laryngectomy was considered as early procedure, while when the surgeon carried out the total laryngectomy after 6 months of follow-up, the salvage laryngectomy was considered as late. Note that positive margins (R1) were followed before decision of total laryngectomy according to the difficulties to have reliable histopathological findings in radiation tissues.

  1. I can not understand the terms of “late” and “early”. You should mention in material and methods.

 Early laryngectomy consisted of TL carried out in the 6 months after the SPL, while late = >6 months of follow-up.

We specified that in the methods: p.2, line 87: “The total laryngectomy was considered in patients with i) positive margins at the post-SPL histopathological analysis, ii) post-operative chondronecrosis or non-functional larynx, or iii) for local recurrence. If the salvage total laryngectomy was performed in the 6-month post-SPL follow-up, the total laryngectomy was considered as early procedure, while when the surgeon carried out the total laryngectomy after 6 months of follow-up, the salvage laryngectomy was considered as late. Note that positive margins (R1) were followed before decision of total laryngectomy according to the difficulties to have reliable histopathological findings in radiation tissues.”

To improve the clearty of the manuscript, we re-specified in the result section: p.4, first paragraph, line 132: “Two patients with chondronecrosis, two patients with positive margins (R1) and one patient with non-functional larynx were treated with early total laryngectomy (<6 month post-SPL). Late total laryngectomy (>6 months of follow-up) was carried out in 5 patients.

  1. You should write details about the contents of the re-intervention.

We provided additional information for the indications and reasons of re-intervention (total laryngectomy) in Appendix 1. You can find in this table, after the TL (total laryngectomy column; with late vs early) the reasons: R1=positive margin and progression during the follow-up; CN=chondronecrosis; recurrence.

As stated below, we added information about the total laryngectomy indications.

As requested by the second reviewer, we also provided information about the neck dissection indications in method, p.4, line 95: “The neck dissection was proposed by the multidisciplinary oncological team in patients with suspicion of positive nodes at the tomodensitometry even if the cTNM classification reported cN0 (<1cm diameter).”

Minor comments

  1. In table 1, is the number of horizontal hemi-laryngectomy 3?

Yes sorry, it is a mistake. We changed 2 in 3: p.3, Table 1, line 120.

Reviewer 2 Report

The authors analyzed the functional and oncological outcome of patients who underwent salvage partial laryngectomy after failure of primary (chemo)radiotherapy.

These are my remarks:

1. The author should specify the primary treatment modality in detail (radiation dose and chemotherapeutics).

2. The surgeons performed 6 unilateral and 5 bilateral neck dissections. In what cases?

3. Are there any cases in which postoperative stenosis was established? It would be advisable to use one of the classifications for the functional outcome (for example, the Vienna classification system).

4. It would be interesting if it is possible to show the survival of those patients who underwent STL after SPL?

Otherwise, I consider the topic very interesting and important for head and neck surgeons.

Author Response

Editor in chief

J Clin Med

Mons, Augustus, 2022

Dear Professor,

I’m sending the revised paper entitled: “Salvage Partial Laryngectomy After failed Radiotherapy: Oncological and Functional Outcomes." (by Gigot et al.) which is submitted for publication in J Clin Med.

I thank the reviewers for the relevant comments. We considered all of them.

Reviewer 2:

The authors analyzed the functional and oncological outcome of patients who underwent salvage partial laryngectomy after failure of primary (chemo)radiotherapy.

These are my remarks:

  1. The author should specify the primary treatment modality in detail (radiation dose and chemotherapeutics).

We added the requested data. Precisely, you can see in Appendix 1 that we added the tobacco status prior to salvage surgery, the alcohol status, the RT doses of initial treatment and the use of chemotherapy in the initial treatment.

You can find the new data in:

-Appendix 1, p.7, line 1, column 2, 3, 5, 6.

-Result section: p.3, setting and patient, line 112: “Eighteen patients (90%) reported a history of tobacco consumption prior to the initial radiation treatment, while 14/16 (88%) had a history of chronic alcohol consumption (> 3 IU/day). The radiotherapy doses of initial treatment are reported in Appendix 1. Two patients had chemoradiation prior to salvage surgery (cisplatin).”

  1. The surgeons performed 6 unilateral and 5 bilateral neck dissections. In what cases?

In case of suspicion of node invasion at the tomodensitometry. In all cases, the cTNM classification was N0 because the nodes were <1cm diameter (criteria of classification). However, the multidisciplinary oncological team supported the realization of functional neck dissection according to the radiologist who suspected nodes <1 cm.

We specified in method, p.4, line 95: “The neck dissection was proposed by the multidisciplinary oncological team in patients with suspicion of positive nodes at the tomodensitometry even if the cTNM classification reported cN0 (<1cm diameter).”

  1. Are there any cases in which postoperative stenosis was established? It would be advisable to use one of the classifications for the functional outcome (for example, the Vienna classification system).

No, we did not observe stenosis over the 5-year follow-up.

  1. It would be interesting if it is possible to show the survival of those patients who underwent STL after SPL?

We checked the medical record of these patients as requested and we added the information in results: “Note that 3/6 patients who underwent salvage total laryngectomy were alive at the end of the follow-up period (5 patients were lost of follow-up). In SPL group, 5/10 patients were alive at the end of the follow-up period (Appendix 1).”

Otherwise, I consider the topic very interesting and important for head and neck surgeons.

Thank you.

Thanking you in advance for your attention, I remain,

Best regards,

Jérôme R. LECHIEN, M.D.,Ph.D., M.S.,

Head and Neck surgery, Laboratory of Anatomy and Cell Biology, Faculty of Medicine, University of Mons (UMONS), Avenue du Champ de mars, 6, B7000 Mons, Belgium, Jerome.Lechien@umons.ac.be

Telephone: +32 65 37 35 84

Fax: +32 65 37 31 42

Round 2

Reviewer 1 Report

comments

1.     As I pointed out previously, this surgical methods and selection of partial laryngectomy depend on the skill of surgeon and the status of patients. For future reference, please provide a flow chart of the ratio of total laryngectomy versus partial laryngectomy following radiation therapy as salvage surgery for laryngeal cancer at your institution. As a result, readers may refer to this result as a guide in making surgical choices.

Author Response

Thank you for the comment.

In sum, we proposed total laryngectomy for all cT4 stage (N=38 patients from 1998 – 2018). For the rest (cT1-T3), as developed in the paper we carried out partial laryngectomy.

We propose this Chart flow as figure 1 to improve the understanding of the paper as proposed by the reviewer.
